

# Mirror stimulation in Eurasian jays (*Garrulus glandarius*)

Luigi Baciadonna[1,2,3,*], Francesca M. Cornero[1,*], Nicola S. Clayton[1] and Nathan J. Emery[2]

[1] Department of Psychology, University of Cambridge, Cambridge, United Kingdom
[2] Biological and Experimental Psychology, School of Biological and Chemical Sciences, Queen Mary University of London, London, United Kingdom
[3] Department of Life Sciences and Systems Biology, University of Turin, Turin, Italy
[*] These authors contributed equally to this work.

## ABSTRACT

Mirror exposure elicits a wide range of behavioral responses, some of which have been considered as part of possible evidence of mirror self-recognition (MSR). These responses can range from social behaviors, indicating that an animal considers its own reflection as a conspecific, to mirror-guided and self-directed actions. Evidence of MSR has been found categorically in only a few species, such as in magpies, chimpanzees, horses, and elephants. Evidence in corvids is currently debated due to inconsistent findings. In this study, we investigated the reaction of Eurasian jays when presenting them with three mirror-stimulation tasks. Based on the overall behavioral patterns across these three tasks, conclusions about birds' understanding of a reflective surface, and their perception of the reflection as either themselves or as a conspecific, appear premature. We highlight how the high neophobia of corvids and other methodological constraints might have hindered the likelihood to approach and explore a mirror, preventing the emergence of behaviors typically associated with MSR. Furthermore, we discuss how motivational factors, methodological constraints and species differences should be considered when interpreting behavioral responses to mirrors.

## INTRODUCTION

Studies of comparative cognition in which a mirror is used generally have one or both of two main objectives: the first is to investigate the presence or absence of mirror self-recognition (MSR) by applying standard MSR tests. The second objective is to investigate whether some inability to process mirrored information might explain a failure to pass the tests developed to explore self-recognition, and usually investigates the capacity for instrumental use of mirrors. With regard to the first objective, the mark-test (*Gallup, 1970*) has become the predominant method of systematically investigating mirror self-recognition abilities in animals. The typical assumption of this test is that MSR is demonstrated when an animal touches or attempts to remove a real mark on their body significantly more than a sham mark (or more frequently than comparable regions of the face without the mark), and more often in the presence of the mirror than not, after a period of exposure to a mirror.

Corresponding author
Luigi Baciadonna,
luigi.baciadonna@qmul.ac.uk

Although the ecological and methodological validity of the mark-test has been questioned (*Heyes, 1994*; *Heyes, 1995*; *De Veer & Van Den Bos, 1999*; *Van den Bos, 1999*; *De Waal, 2019*; *Vonk, 2019*), several species such as great apes (for a review see *Anderson & Gallup (2015)*), dolphins (*Reiss & Marino, 2001*; *Loth et al., 2022*), elephants (*Plotnik, De Waal & Reiss, 2006*), corvids (*Prior, Schwarz & Güntürkün, 2008*; *Dally, Emery & Clayton, 2010*; *Clary & Kelly, 2016*; *Buniyaadi, Taufique & Kumar, 2020*), horses (*Baragli et al., 2021*) and cleaner wrasse (*Kohda et al., 2019*; *Kohda et al., 2022*) have shown behavioral patterns in line with passing this test.

With regard to the second objective, instrumental mirror use, tasks typically involve mirror image stimulation, mirror-triggered search, mirror-mediated object discrimination, mirror-mediated spatial locating, and mirror-guided reaching (*Menzel, Savage-Rumbaugh & Lawson, 1985*; *Povinelli, 1989*; *Pepperberg et al., 1995*). The mirror image stimulation is also often carried out before the mark-test: subjects are exposed to their own reflection while their behaviors are observed. Animals that may possess MSR, over time, start exploring the mirror, initially performing social and contingent behaviors and, eventually, their behavior in front of the mirror becomes self-directed, opening up the possibility that they perceive the reflection as their own (*Gallup, 1970*). As such, self-directed behaviors in absence of any formal training have been considered a prerequisite to subsequently carry out a mark-test (*Gallup, 1970*; *Plotnik, De Waal & Reiss, 2006*; *De Waal, 2019*). The mirror-triggered search is a basic task in which an animal can be motivated to search for, and possibly find, a food item not directly visible, with the aid of a mirror (*Anderson, 1986*; *Povinelli, 1989*; *Pepperberg et al., 1995*; *Broom, Sena & Moynihan, 2009*; *Howell & Bennett, 2011*; *Gieling et al., 2014*; *Wang et al., 2020*). In the mirror-mediated object discrimination task, the subject is able to locate and approach positive stimuli, and avoid negative stimuli, with the use of a mirror, and even when the stimuli are located in a new position (*Menzel, Savage-Rumbaugh & Lawson, 1985*; *Pepperberg et al., 1995*). In the mirror-mediated spatial locating task, subjects have to find rewards, visible in the mirror, in one of multiple locations available (*Menzel, Savage-Rumbaugh & Lawson, 1985*; *Anderson, 1986*; *Povinelli, 1989*; *Pepperberg et al., 1995*; *Medina et al., 2011*). In the mirror-guided reaching task, subjects can only reach the reward by monitoring and adjusting their own movements while looking at the mirror (*Menzel, Savage-Rumbaugh & Lawson, 1985*; *Anderson, 1986*; *Itakura, 1987*; *Povinelli, 1989*; *Baciadonna et al., 2021a*). These tasks require different levels of processing of the information gained by the mirror, from simply using the mirror as a trigger to start searching for something rewarding, to grasping the correlation between an object and its reflection, to a more advanced understanding of the correspondence between the location of the object in space and its reflected image, to finally being able to connect the subject's own movements to those reflected in the mirror (*Pepperberg et al., 1995*). While studies investigating MSR are abundant, studies where the main focus of attention is to explore animal's ability to process mirrored information are less prevalent.

When considering mirror studies involving avian subjects, several have investigated MSR, in particular with large-brained species such as parrots and corvids (*Derégnaucourt & Bovet, 2016*; *Brecht & Nieder, 2020*; *Baciadonna et al., 2021b*). Three studies claim to find evidence of MSR in corvids (*Prior, Schwarz & Güntürkün, 2008*; *Clary & Kelly, 2016*;

*Buniyaadi, Taufique & Kumar, 2020*). However, the claims about MSR in birds are highly debated, with later studies failing to replicate initial findings (*Soler et al., 2020*; *Parishar, Mohapatra & Iyengar, 2021*), and other studies reporting negative results (carrion and hooded crows, *Vanhooland, Bugnyar & Massen, 2020*; *Brecht, Müller & Nieder, 2020*; azure-winged magpies, *Wang et al., 2020*; great tits, *Kraft et al., 2017*; keas and Goffin's cockatoos, *Van Buuren et al., 2018*; ravens, *Baciadonna et al., 2022*; *Vanhooland et al., 2023*). In the studies in which a mark-test was conducted, birds were also, as would be expected, presented with a mirror stimulation test in which their behaviors towards the mirror itself were recorded. At a general level, studies conducted on captive jungle and New Caledonian crows suggest that these birds considered their mirrored image as a conspecific, and did not display self-directed behavior during the mirror image stimulation (*Kusayama, Bischof & Watanabe, 2000*; *Medina et al., 2011*). Carrion, hooded crows, ravens and azure-winged magpies displayed exploratory behaviors, social behaviors and to some extent contingent behaviors, but none displayed significant mark-directed behaviors (*Brecht, Müller & Nieder, 2020*; *Vanhooland, Bugnyar & Massen, 2020*; *Baciadonna et al., 2022*). However, in only few studies (*Pepperberg et al., 1995*; *Kusayama, Bischof & Watanabe, 2000*; *Taylor et al., 2010*; *Medina et al., 2011*; *Wang et al., 2020*; *Baciadonna et al., 2021a*) were birds exposed to an image stimulation and/or to some instrumental mirror-use tasks without being tested with the mark-test. African grey parrots have been successful at mirror-mediated object discrimination, mirror-triggered search, and mirror-guided spatial locating (*Pepperberg et al., 1995*), New Caledonian crows solved a mirror-guided spatial locating task (*Medina et al., 2011*) and azure-winged magpies failed a mirror-guided spatial locating task (*Wang et al., 2020*), but the mirrored image of a treat did trigger search efforts. Generally, avian species exposed to mirror-image stimulation react to their reflected image socially, but a few of the species did not show social responses (or these decreased over habituation), and possible contingency checking occurred (*Prior, Schwarz & Güntürkün, 2008*; *Van Buuren et al., 2018*; *Vanhooland, Bugnyar & Massen, 2020*; *Buniyaadi, Taufique & Kumar, 2020*; *Vanhooland et al., 2023*). Instances of self-examination are extremely rare, but one parrot may have done so by examining the bottom of its foot in the mirror and real life simultaneously (*Pepperberg et al., 1995*). It has also been noted that exposing birds to a horizontal, rather than vertical, mirror appears to decrease the amount of social responses or facilitate habituation (*Pepperberg et al., 1995*; *Kusayama, Bischof & Watanabe, 2000*; *Van Buuren et al., 2018*).

To our knowledge, Eurasian jays, a corvid species, have not been formally tested in the mark-test. However, in a recent study, a sample of Eurasian jays were tested in a mirror-guided reaching task using a modified version of the horizontal string-pulling task. Although four birds learned to pull the correct string when they could see the food directly, none used the reflected information to successfully retrieve the reward (*Baciadonna et al., 2021a*). The results of the mirror-guided reaching task do not necessarily exclude the possibility that these birds could still be able to use the mirror instrumentally in less demanding tasks, such as mirror triggered search, mirror-mediated object discrimination, or mirror-mediated spatial locating tasks. More importantly, to date, Eurasian jays' reaction toward the mirror during a mirror image stimulation have not been described. In addition,

based on their cognitive abilities, Eurasian jays are a very interesting model to study their natural response to mirrors, and to explore possible precursors to MSR (*Baciadonna et al., 2021b*).

In the present study, we explored the responses of Eurasian jays towards three mirror tasks: a mirror preference task (Task 1), a mirror preference task with varying food quality (Task 2), and a vertical *vs.* horizontal mirror habituation task (Task 3). These tasks were progressively intended to both assess the jays' responses to their mirrored image as well as to encourage them to increasingly explore their mirror image, with the possibility that if there was an eventual display of behaviors that might indicate emerging MSR abilities, this would have then warranted subsequent presentation of a formal mark-test. However, given the results observed in Tasks 1-3, this was ultimately not performed. In Task 1, jays were presented with a choice of sitting on a perch and eating either in front of a mirror or in front of a medium-density fiberboard (MDF, a non-reflective surface), with two equivalent amounts of food of equal palatability one in front of each surface. The purpose of Task 1 was to assess whether the birds displayed a preference for the mirrored surface compared to the non-reflective surface or vice-versa, and thus it was to provide preliminary evidence as to how the birds may perceive their reflected image. A preference for the mirror may have indicated either a social response towards it (there is some evidence that corvids consume more food when in the presence of conspecifics; *Dally, Clayton & Emery, 2008*) or an interest in assessing their own image, depending on concurrent behaviors displayed. A preference for the non-reflective surface may have indicated an avoidance of a threatening conspecific or of the mirror stimuli. On the other hand, the absence of a preference for one of the two conditions may indicate that both surfaces were not considered to be either threatening or particularly interesting. Task 2 was similar to Task 1, but a bowl with a more-palatable food was placed in front of the mirror and the birds' normal daily food was placed in a bowl in front of the wooden panel. Additionally, birds had to take the time to remove a layer of cling film from the bowl with more-palatable food and from the bowl with normal daily food to access the food. Task 2 was designed after Task 1 was presented to investigate whether the presence of a more-palatable food could increase exploratory behaviors towards the mirror panel by encouraging birds to overcome the lack of interest they initially displayed towards the mirror in Task 1. Encouraging the birds to spend time in front of the mirror by providing a more-palatable food that was slow to access may have given them time to notice and confront their own reflection as they worked to obtain the food, which they could not do if they simply avoided it or took the food quickly and flew away. In Task 3, we additionally assessed the birds' latency to approach and collect a food reward when it required the jays to land close to one of two different mirror configurations (vertical or horizontal) at a time, comparing their latency to approach when later presented with the same configurations of wooden boards rather than reflective surfaces. Task 3 was designed after Task 2 was presented, to provide the birds with further experience with a mirror in which they were required to approach the reflection of their own head and face to obtain the food (because their lack of interest in their mirrored reflection continued in Task 2), as well as to examine whether the physical configuration of a mirrored surface affected the birds' behavior towards it. It is possible that a horizontally-presented mirror

may be more naturalistic to the birds (such as a reflection from a water source) and may thus have been perceived as less unusual or threatening (whereas a vertical reflection is more likely to be similar to how they would encounter a conspecific, rather than their own reflection; *Pepperberg et al., 1995*; *Kusayama, Bischof & Watanabe, 2000*; *Derégnaucourt & Bovet, 2016*), in which case the birds may behave differently towards a horizontal mirror than towards the vertical mirrors to which they had been exposed, by approaching to retrieve rewards faster from a horizontal mirror.

## MATERIALS AND METHODS

### Location, subjects, housing condition and animal ethics

Eight adult Eurasian jays housed at the Comparative Cognition Laboratory at the Sub-Department of Animal Behavior, University of Cambridge in Madingley, United Kingdom were tested. The jays were housed in their social groups in two large outdoor aviaries (20 × 6 × 3 m): Five in Aviary I (Caracas, Lisbon, and Lima, males; Wellington and Washington, females, 13 years old) and three in Aviary II (Romero and Hoy, males; Hunter, female, 14 years old). However, only seven jays were tested in Tasks 1 and 2: Lisbon was not included because he initially refused to come inside the testing compartment, but then started spontaneously entering for Task 3, and so was included then. Smaller indoor testing compartments (3 × 1 × 2 m) connected to the aviary by hatch doors (0.5 × 0.5m) were used for testing. Subjects participated voluntarily. The testing compartments contained two suspended platforms (1 × 1 m) approximately 1 m from the ground, where the birds could walk or land to rest. During testing sessions, which occurred once daily and lasted 15 min per subject, each individual was physically and visually isolated from other jays. Birds were food deprived for an hour before testing (birds were never food deprived for more than 4 hours/day and were never water deprived). Outside of testing, birds were fed a maintenance diet of soaked cat biscuits, vegetables, seeds, fruit, and hard-boiled eggs. The jays were hand-raised by licensed breeders and had since lived in laboratory settings. Furthermore, these jays had previously participated in different experiments (*e.g.*, *Shaw & Clayton, 2014*; *Legg, Ostojić & Clayton, 2016*; *Ostojić et al., 2016*; *Amodio et al., 2021*). They had also had exposure to mirrors during previous mirror-stimulation experiments: these included the placement of a mirror inside their aviaries for two weeks and a mirror-guided string-pulling study (*Baciadonna et al., 2021a*). All experiments were approved by the University of Cambridge (ZOO63/19) and followed Home Office Regulations and the ASAB's Guidelines for the Treatment of Animals in Behavioral Research and Teaching. At the end of the study, jays were kept in their aviaries in their respective social groups.

### Experimental set-up and procedure

#### Task 1: mirror preference task

In this task, each jay was moved into an indoor testing compartment and had the chance to approach and retrieve an identical reward (eight peanuts and eight macadamia nuts) from the front of either a wooden or mirror panel; both surfaces were present at the same time. On the longer side of the indoor testing compartment, a wooden (MDF) panel and a mirror panel of the same size (45 × 45 cm) were suspended vertically from the wire

mesh using a metal hook; the sides on which the mirror and wood panels were placed were counterbalanced per subject. A small food container and a small resting perch (15 cm long and 20 cm distant from each panel) were fixed at the base of each panel. Each food container had eight half macadamia nuts and eight half peanuts. In total, the jays had 16 fifteen-minute sessions each (one session per day: total time 240 min).

*Task 2: mirror preference task with varying food quality*
In this task, each jay was moved into an indoor testing compartment and had the opportunity to approach and retrieve either a more-desirable food (20 waxworms) from the front of a mirrored panel, or a less-desirable food (dried cat biscuits from their daily maintenance diet) from the front of a wooden panel (waxworms are used as experimental treats for the birds in most of our experiments), both surfaces were present at the same time. Furthermore, to reduce the speed of food consumption and increase the likelihood that the birds would explore the surfaces more, each food bowl was covered with cling-film that needed to be removed to retrieve the food. Before starting Task 2, a habituation phase (one session per day) was performed to allow the birds to learn how to remove a cling-film lid placed on top of a food bowl (placed in the center of the wooden platforms, without any corresponding test surfaces) to retrieve a small portion of mealworms. To move on to Task 2, subjects had to successfully remove the cling-film from the bowl in four consecutive sessions. Once they were moved on to Task 2, the jays had eight fifteen-minute sessions each (one session per day: total time 120 min).

*Task 3: vertical vs. horizontal mirror habituation task*
In this task, each jay was moved into an indoor testing compartment and could approach and retrieve a food reward from the center of a panel, either vertical or horizontal, and either mirrored or wooden. Only one surface and position configuration at one time was placed in the testing compartment, resulting in four different conditions being tested. Subjects experienced first a vertical mirror (VM) and a horizontal mirror (HM) condition and, subsequently, a vertical wood (VW) and a horizontal wood (HW) condition. All subjects experienced each mirror condition before either of the wooden conditions. The wooden conditions were added once the hesitancy of the birds to approach the mirrors was noted, in order to present a point of comparison with a material they were more familiar with. The placement of surfaces was otherwise randomized between subjects, so that half of the birds experienced the vertical conditions before the horizontal ones, and vice-versa. Mirrors and wooden surfaces were flat 30 × 30 cm squares. For the vertical condition, the surface hung from the compartment's mesh side using a metal hook and wire and rested on the wooden suspended platform so that a jay could walk directly up to it. For the horizontal condition, the surface was placed flat on the wooden platform inside the compartment, equidistant from both mesh sides. Glued at the center of each surface was a small, transparent Plexiglas tube, approximately one cm in diameter and height, where a live waxworm would be baited at the beginning of each session. The center position of the waxworm would force the birds to approach, and hopefully see, their own mirrored head and face when retrieving a worm (in the mirror conditions). When a bird approached the surface and successfully retrieved the worm, the tube was then immediately re-baited, until

**Table 1   Ethogram used for Mirror Preference Task (Task 1), and Mirror Preference Task with Varying Food Quality (Task 2).**

| Behavioral category | Behavior with description |
|---|---|
| Exploration | Food taken: number of food item taken either from the mirror of the wooden panel |
| | Duration of looking: duration of looking towards the mirror or wood panel with the body and head facing them |
| | Occurrences of looking: occurrences of looking towards the mirror or wood panel with the body and head facing them |
| Social behavior | Aggressive/defensive: occurrence of jumps toward the mirror or wood panel usually with claws up and wings movements |
| Contingent behavior | Head movement: occurrence of repetitive movement of the head (*e.g.*, left/right or up/down) when in front of the mirror or wood panel |
| Self-directed behavior | Preening: duration of preening when the beak grooms parts of the body when facing the mirror or the wooden panel |

a bird had successfully retrieved a worm 20 times (either within one 15 min session, or over multiple sessions). If a bird did not pass a condition, testing on that condition ended after the bird had failed to retrieve any worm over five consecutive 15-minute sessions.

## Video coding

A digital video camera (GoPro Hero4) was used to record all test sessions. For Tasks 1 and 2, the videos were scored using Behavioral Observation Research Interactive Software (BORIS v. 7.7.3; *Friard & Gamba, 2016*). The ethogram used, largely based on the current literature (*Prior, Schwarz & Güntürkün, 2008*; *Soler et al., 2020*; *Vanhooland, Bugnyar & Massen, 2020*) included, at the initial stage, the following behavioral categories: exploration (food taken, duration and occurrences of looking), social behavior, contingent behavior and self-directed behavior (Table 1). However, considering jays' overall low engagement with the mirror (0.10 occurrences per minute for Task 1 and 0.008 occurrences per minute for Task 2), only the most conspicuous behaviors displayed were investigated statistically (Tables 2 and 3). For Tasks 1 and 2, the amount of food taken either from the mirror or the wooden panel conditions and the duration (sec) and occurrences of looking (body and head facing either the mirror or the wooden panel) were scored. LB coded all the videos for Tasks 1 and 2. Twenty per cent of the videos randomly chosen were scored by a second independent observer (MM). The interclass correlation coefficient calculated for all the behaviors analysed statistically was: 0.98 for duration of looking and 0.91 for occurrences of looking. For Task 3, the latency, *i.e.,* the time elapsed between a bird successfully retrieving the worm from the surface and the moment in which the experimenter's arm was removed from the compartment after baiting the Plexiglas tube, was calculated. For Task 3, the latency was scored directly during testing: the stopwatch was controlled by a second experimenter (FMC), while the first experimenter (LB) was in charge of baiting the Plexiglas tube.

## Statistical analyses

R version 3.6.1 (*R Development Core Team, 2020*) was used for all statistical analyses. For Task 1, a model was calculated for each of the following dependent variables: food taken and occurrences of looking. Condition (mirror, wood), Session (1-16), Aviary (group that each bird belonged to, with two levels), and Sex (female, male) were included as fixed factors for all models performed for Task 1 to control for potential differences. For

**Table 2  Individual responses in the Mirror Preference Task (Task 1).**

| | Subject | | | | | | | | | | | | | |
| | Romero | | Hoy | | Lima | | Washington | | Caracas | | Wellington | | Hunter | |
| Condition Behavior | Mirror | Wood | Mirror | Wood | Mirror | Wood | Mirror | Wood | Mirror | Wood | Mirror | Wood | Mirror | Wood |
|---|---|---|---|---|---|---|---|---|---|---|---|---|---|---|
| Food taken | 16 | 32 | 0 | 13 | 0 | 21 | 3 | 4 | 1 | 12 | 0 | 1 | 1 | 6 |
| Duration of looking (s) | 357.08 | 250.14 | 133.80 | 95.79 | 65.71 | 328.94 | 119.49 | 72.41 | 169.08 | 386.97 | 106.62 | 17.49 | 51.62 | 91.70 |
| Occurrence of looking | 38 | 43 | 22 | 14 | 18 | 25 | 37 | 18 | 84 | 85 | 8 | 3 | 13 | 16 |
| Social behavior | 1 | 0 | 0 | 0 | 0 | 0 | 0 | 0 | 4 | 0 | 0 | 0 | 0 | 0 |
| Contingent | 17 | 0 | 4 | 0 | 0 | 0 | 0 | 0 | 0 | 0 | 0 | 0 | 0 | 0 |
| Self-directed | 0 | 0 | 0 | 0 | 0 | 0 | 0 | 0 | 0 | 0 | 0 | 0 | 0 | 0 |

**Table 3  Individual responses in the Mirror Preference Task with Varying Food Quality (Task 2).**

| | Subject | | | | | | | | | | | | | |
| | Romero | | Hoy | | Lima | | Washington | | Caracas | | Wellington | | Hunter | |
| Condition Behavior | Mirror | Wood | Mirror | Wood | Mirror | Wood | Mirror | Wood | Mirror | Wood | Mirror | Wood | Mirror | Wood |
|---|---|---|---|---|---|---|---|---|---|---|---|---|---|---|
| Food taken | 142 | 4 | 160 | 17 | 160 | 0 | 158 | 1 | 0 | 0 | 0 | 0 | 0 | 0 |
| Duration looking (s) | 1155.02 | 246.02 | 619.25 | 257.77 | 1255.91 | 72.66 | 781.43 | 64.27 | 109.25 | 57.28 | 0 | 0 | 32.77 | 3.75 |
| Occurrence of looking | 39 | 15 | 16 | 14 | 14 | 11 | 64 | 13 | 10 | 14 | 0 | 0 | 7 | 1 |
| Social behavior | 0 | 0 | 0 | 0 | 0 | 0 | 0 | 0 | 0 | 0 | 0 | 0 | 0 | 0 |
| Contingent | 0 | 0 | 0 | 0 | 1 | 0 | 0 | 0 | 0 | 0 | 0 | 0 | 0 | 0 |
| Self-directed | 0 | 0 | 0 | 0 | 0 | 0 | 0 | 0 | 0 | 0 | 0 | 0 | 0 | 0 |

Task 1, the Generalized Linear Mixed Models using Template Model Builder (glmmTMD package; *Brooks et al., 2017*) was used because the dependent variables food taken (Poisson distribution) and occurrences of looking violated the normality assumption, as well as due to the high frequency of occurrences of zero-values. The dependent variable, duration of looking, was analyzed using non-parametric methods for Task 1. Wilcoxon signed-rank tests were conducted to identify any significant differences between the duration of looking for each condition.

For Task 2, generalized linear mixed models were calculated using the lme4 package (*Bates et al., 2015*). For each of the following dependent variables: food taken (binomial distribution), occurrences, and duration of looking. Condition (mirror, wooden panel), Session (1-8), Aviary (group that each bird belonged to, with two levels), and Sex (female, male), were included as fixed factors for all models performed for Task 2 to control for potential differences.

For Task 3, the latency to approach the food was analyzed. The data obtained violated the normality assumption, and thus a non-parametric approach was employed for analysis. A Fisher's exact test was used to compare the number of birds that passed or failed each condition. A Friedman test was conducted to identify any significant differences between conditions in both latency to approach and the average number of sessions required to pass each condition.

**Table 4   Results of the GLMM TDM showing which variables affected the food taken in Mirror Preference Task (Task 1).**

|  | Estimate | Standard error | z | p-value |
|---|---|---|---|---|
| Intercept | −1.13 | 0.57 | −1.96 |  |
| Condition | 1.02 | 0.36 | 2.76 | 0.003 |
| Session | −0.01 | 0.02 | −0.64 | 0.51 |
| Aviary | −0.35 | 0.32 | −1.10 | 0.30 |
| Sex | 1.71 | 0.39 | 4.35 | 0.001 |

**Table 5   Results of the GLMM showing which variables affected the occurrences of looking Mirror Preference Task (Task 1).**

|  | Estimate | Standard error | z | p-value |
|---|---|---|---|---|
| Intercept | −0.06 | 0.38 | −0.15 |  |
| Condition | 0.11 | 0.10 | 1.12 | 0.26 |
| Session | 0.02 | 0.01 | 2.08 | 0.035 |
| Aviary | 0.33 | 0.36 | 0.93 | 0.38 |
| Sex | 0.85 | 0.37 | 2.25 | 0.042 |

For all GLMM models, the significance of the full model was established by comparing this model with the model that included only the random factor (null model) using a likelihood ratio test. Model fit and over-dispersion were checked using the DHARMa 0.3.3.0 package (*Harting, 2020*). The *p*-value of each factor was derived using the "drop1" function (*Barr et al., 2013*). Also, the subjects' identity was included as a random factor to control for repeated measurements of the same subject in all models performed.

## RESULTS

### Task 1: mirror preference task

When investigating which variables affected the food taken from either surface, it was found that the full model differed significantly from the null model ($AIC_{null} = 366.97$ *vs* $AIC_{full} = 353.38$; GLMM: $\chi^2 = 21.59$, $df = 4$, $p < 0.0001$). The fixed factor Condition was significant (Table 4); jays took more food from in front of the wooden panel (Mean ± SE = 0.79 ± 0.25) compared to from in front of the mirror panel (Mean ± SE = 0.18 ± 0.13; Fig. 1A). The fixed Sex factor was also significant (Table 4), with male jays taking more food (Mean ± SE = 0.74 ± 0.25) than females (Mean ± SE = 0.15 ± 0.06), across conditions. The other fixed factors included in the model were not significant (Table 4). When investigating which variables affected the occurrences of looking, it was found that the full model differed significantly from the null model ($AIC_{null} = 831.06$ *vs* $AIC_{full} = 828.03$; GLMM: $\chi^2 = 11.02$, $df = 4$, $p = 0.026$). The fixed Sex factor was significant (Table 5), with male jays looking more often (Mean ± SE = 2.57 ± 0.95) than females (Mean ± SE = 0.98 ± 0.39), across conditions. The factor Session was also significant (Table 5). Overall, across the sessions, jays looked either at the mirror or the wooden panel on average 1.89
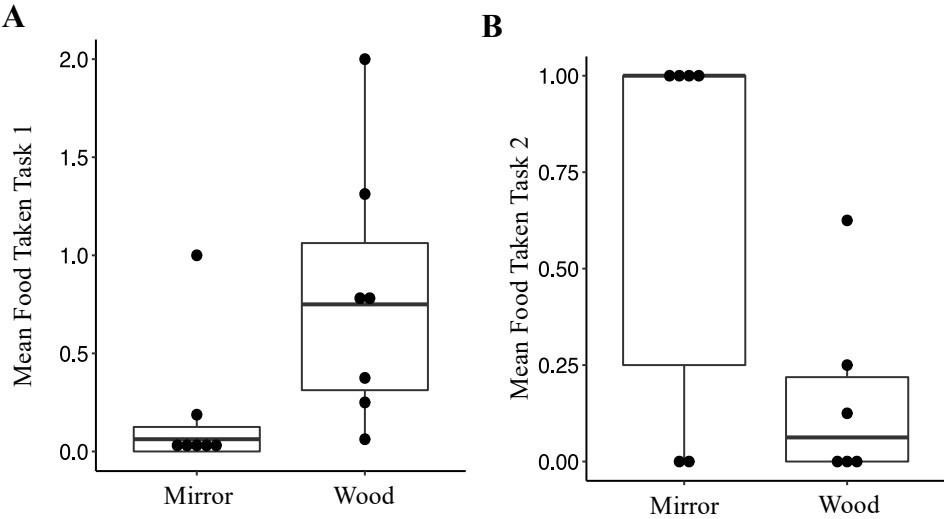

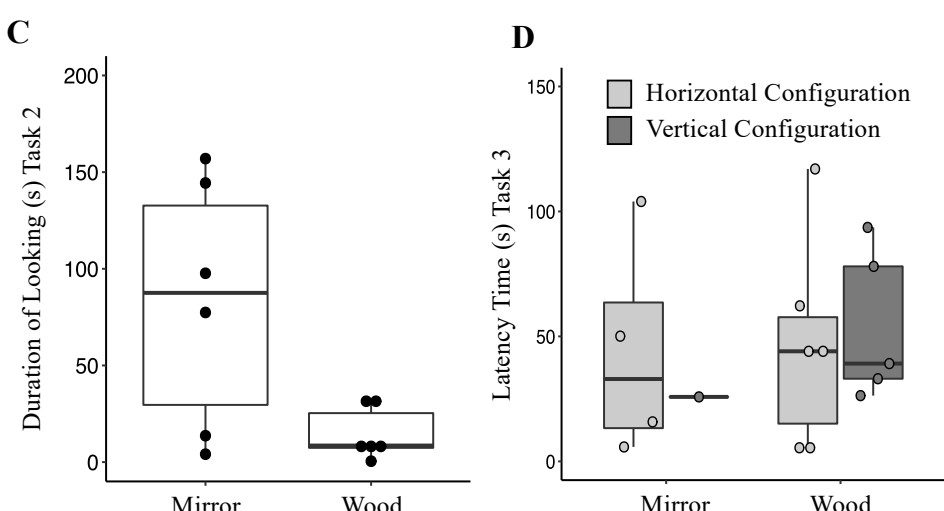

**Figure 1** **(A) Mean of food taken in front of the mirror and the wooden panel during Mirror Preference Task (Task 1); (B) Mean of food taken in front of the mirror and the wooden panel during Mirror Preference Task with Varying Food Quality (Task 2); (C) Total duration of looking time towards the mirror and the wooden panel during the Mirror Preference Task with Varying Food Quality (Task 2); (D) Latency time to approach and retrieve a waxworm across the four combinations presented to the jays in Vertical *vs.* Horizontal Mirror Habituation Task (Task 3); Mirror, horizontal (MH) and vertical (MV); Wood, horizonal (WH) and vertical (WV).** Box plot: the horizontal line shows the median, the box extends from the lower to the upper quartile and the whiskers to the interquartile range above the upper quartile (max) or below the lower quartile (min); solid circles indicate each individual jay.

± 0.20 times. In the last two sessions, the occurrences of looking increased (Session 15, Mean ± SE = 3.21 ± 1.70; Session 16, Mean ± SE = 2.78 ± 1.80). The other fixed factors included in the model were not significant (Table 5). A Wilcoxon signed-rank test showed

**Table 6   Results of the GLMER showing which variables affected the food taken in Mirror Preference Task with Varying Food Quality (Task 2).**

|  | Estimate | Standard error | z | p-value |
|---|---|---|---|---|
| Intercept | 0.42 | 5.71 | 0.075 | |
| Condition | −7.74 | 2.94 | −2.62 | <0.0001 |
| Session | −0.03 | 0.19 | −0.19 | 0.84 |
| Aviary | −1.60 | 4.86 | −0.33 | 0.74 |
| Sex | 3.85 | 5.47 | 0.70 | 0.46 |

**Table 7   Results of the GLMM showing which variables affected occurrences of looking in Mirror Preference Task with Varying Food Quality (Task 2).**

|  | Estimate | Standard error | z | p-value |
|---|---|---|---|---|
| Intercept | 1.31 | 0.58 | 2.24 | |
| Condition | −0.56 | 0.17 | −3.32 | 0.004 |
| Session | −0.13 | 0.03 | −4.17 | 0.007 |
| Aviary | 0.36 | 0.54 | 0.67 | 0.52 |
| Sex | 0.05 | 0.58 | 0.09 | 0.62 |

**Table 8   Results of the GLMER showing which variables affected the duration of looking in Mirror Preference Task with Varying Food Quality (Task 2).**

|  | Estimate | Standard error | t | p-value |
|---|---|---|---|---|
| Intercept | 77.85 | 25.36 | 3.06 | |
| Condition | −67.74 | 8.22 | −8.23 | <0.0001 |
| Session | −3.70 | 1.79 | −2.06 | 0.041 |
| Aviary | 0.54 | 23.68 | 0.02 | 0.98 |
| Sex | 31.38 | 25.12 | 1.24 | 0.23 |

that the duration of looking either at the mirror (Mean $\pm$ SE = 8.95 $\pm$ 2.41 s) or at the wooden panel (Mean $\pm$ SE = 11.10 $\pm$ 3.37 s) did not differ ($V = 1381.5$, $p = 0.67$).

## Task 2: mirror preference task with varying food quality

Six out of seven birds managed to remove the cling-film during the habituation phase (range 4-12 sessions). One subject was excluded (Wellington) because she never managed to remove the cling-film and retrieve the food in four consecutive sessions. When investigating which variables affected the food taken, it was found that the full model differed significantly from the null model (AIC$_{null}$ = 105.36 *vs* AIC$_{full}$ = 60.34; GLMM: $\chi^2 = 53.01$, $df = 4$, $p < 0.001$). The fixed factor Condition was significant (Table 6); jays took more food in front of the mirror panel (Mean $\pm$ SE = 0.66 $\pm$ 0.21) compared to the wooden panel (Mean $\pm$ SE = 0.16 $\pm$ 0.10; Fig. 1B). The other fixed factors included in the model were not significant (Table 6). When investigating which variables affected the occurrences of looking, it was found that the full model differed significantly from the

null model (AIC$_{null}$ = 422.89 $vs$ AIC$_{full}$ = 405.56; GLMM: $\chi^2$ = 25.33, $df$ = 4, $p$ < 0.0001). The fixed factor Condition was significant (Table 7), in that jays looked more often at the mirror panel (Mean ± SE = 3.12 ± 1.13) compared to the wooden panel (Mean ± SE = 1.41 ± 0.26; Fig. 1C). The factor Session was also significant (Table 7). Overall, the amount of looks across sessions decreased (Session 1, Mean ± SE = 4.41 ± 0.96; Session 8, Mean ± SE = 2.16 ± 1.07). The other fixed factors included in the model were not significant (Table 7). When investigating which variables affected the duration of looking, it was found that the full model differed significantly from the null model (AIC$_{null}$ = 1055.2 $vs$ AIC$_{full}$ = 1008.9; GLMM: $\chi^2$ = 54.31, $df$ = 4, $p$ < 0.0001). The fixed factor Condition was significant (Table 8), with jays looking at the mirror panel for longer (Mean ± SE = 82.36 ± 26.25 s) than the wooden panel (Mean ± SE = 14.61 ± 5.47 s). The factor Session was also significant (Table 8). Overall, the duration of looking across sessions decreased (Session 1, Mean ± SE = 78.09 ± 23.43 s; Session 8, Mean ± SE = 45.01 ± 15.33 s). The other fixed factors included in the model were not significant (Table 8).

## Task 3: vertical *vs.* horizontal mirror habituation task

The latencies to approach the food across the four conditions are represented in Fig. 1D. Only one out of eight jays successfully retrieved the reward 20 times in the VM configuration (Romero; in two sessions). Five out of eight jays successfully obtained the reward 20 times (session range, 2-9) in the HM configuration (Mean ± SE = 75.50 ± 36.07s; Romero, Hoy, Lima, Washington, and Hunter). When tested in the VW configuration, five out of eight subjects (Mean ± SE = 54.02 ± 13.36 s; Romero, Lima, Washington, Hunter, Lisbon) successfully obtained the reward 20 times (session range, 1–5), whereas six (Romero, Lima, Washington, Hunter, Lisbon, Hunter) successfully retrieved the reward (session range, 1-8) in the HW configuration (Mean ± SE = 46.37 ± 16.93 s). Two birds (Caracas and Wellington) did not complete any of the four conditions (Table 9). However, there was not a significant association between the type of configuration and whether the jays succeeded (Fisher's exact test, $p$ = 0.084). The latency to approach the food was compared between three conditions (HM, and VW and HW). One condition, the VM, was excluded because of an insufficient number of observations (only one jay successfully completed the task). The latency to approach the food was not significantly different between conditions (Friedman test: $\chi^2$ = 3.5, $df$ = 2, $p$ = 0.17). The number of sessions required to complete Task 3 also did not differ between conditions (Friedman test: $\chi^2$ = 4.30, $df$ = 2, $p$ = 0.11).

## DISCUSSION

Across the three tasks, Eurasian jays did not display the expected behavioral transition from initial social behaviors to exploration, contingency testing and self-directed behaviors that are typically observed in species in which mirror self-recognition has been reported. During Tasks 1 and 2, only three birds showed contingent behaviors, and none of the birds showed mirror guided self-exploration, which has been considered the main sign that needs to be observed before performing a classic mark-test (De Waal, 2019). Therefore, we did not proceed to designing or conducting a mark-test with these birds. The birds' strong preference for taking the food from in front of the wooden panel suggests, according to

Table 9 **Summary of the jays' performance to approach and retrieve the food during Vertical vs. Horizontal Mirror Habituation Task (Task 3). Plus (+) indicates the instances in which the tested subject approached and retrieved at least once the waxworm during the 15 min allowed in each session. Minus (−) indicates the instances in which the tested subject did not approach and retrieve at least once the waxworm during the 15 min allowed in each session.**

| ID | Position | Condition | Sessions | | | | | | | | | | | | | Tot |
|---|---|---|---|---|---|---|---|---|---|---|---|---|---|---|---|---|
| | | | 1 | 2 | 3 | 4 | 5 | 6 | 7 | 8 | 9 | 10 | 11 | 12 | 13 | |
| Romero | Vertical | Mirror | + | + | | | | | | | | | | | | 20/20 |
| | Horizontal | Mirror | − | − | + | + | | | | | | | | | | 20/20 |
| | Horizontal | Wooden | + | | | | | | | | | | | | | 20/20 |
| | Vertical | Wooden | + | | | | | | | | | | | | | 20/20 |
| Hoy | Horizontal | Mirror | − | − | + | + | + | | | | | | | | | 20/20 |
| | Vertical | Mirror | + | − | − | − | − | − | | | | | | | | 1/20 |
| | Vertical | Wooden | − | − | − | − | − | | | | | | | | | 0/20 |
| | Horizontal | Wooden | + | + | + | | | | | | | | | | | 20/20 |
| Lima | Vertical | Mirror | − | − | − | − | − | | | | | | | | | 0/20 |
| | Horizontal | Mirror | − | + | | | | | | | | | | | | 20/20 |
| | Horizontal | Wooden | + | | | | | | | | | | | | | 20/20 |
| | Vertical | Wooden | − | + | | | | | | | | | | | | 20/20 |
| Washington | Horizontal | Mirror | + | | | | | | | | | | | | | 20/20 |
| | Vertical | Mirror | + | + | − | + | − | − | + | + | − | − | − | − | − | 14/20 |
| | Vertical | Wooden | − | + | | | | | | | | | | | | 20/20 |
| | Horizontal | Wooden | + | | | | | | | | | | | | | 20/20 |
| Caracas | Vertical | Mirror | − | − | − | − | − | | | | | | | | | 0/20 |
| | Horizontal | Mirror | − | + | − | − | − | − | − | | | | | | | 8/20 |
| | Horizontal | Wooden | + | + | − | − | − | − | − | | | | | | | 2/20 |
| | Vertical | Wooden | − | − | − | − | − | | | | | | | | | 0/20 |
| Wellington | Horizontal | Mirror | − | − | − | − | − | | | | | | | | | 0/20 |
| | Vertical | Mirror | − | − | − | − | − | | | | | | | | | 0/20 |
| | Vertical | Wooden | − | − | − | − | − | | | | | | | | | 0/20 |
| | Horizontal | Wooden | − | − | − | − | − | | | | | | | | | 0/20 |
| Hunter | Vertical | Mirror | − | − | − | − | − | | | | | | | | | 0/20 |
| | Horizontal | Mirror | + | + | − | − | − | − | − | + | − | | | | | 20/20 |
| | Horizontal | Wooden | + | + | | | | | | | | | | | | 20/20 |
| | Vertical | Wooden | − | + | + | + | | | | | | | | | | 20/20 |
| Lisbon | Horizontal | Mirror | + | + | − | − | − | − | − | | | | | | | 7/20 |
| | Vertical | Mirror | − | − | − | − | − | | | | | | | | | 0/20 |
| | Vertical | Wooden | − | − | + | + | + | | | | | | | | | 20/20 |
| | Horizontal | Wooden | + | + | + | − | − | + | − | + | | | | | | 20/20 |

our initial prediction, that their reflected image was perceived as unusual, threatening and to be avoided. This initial avoidance reaction towards their reflected image was attenuated by providing more palatable food placed near to the mirror. However, although the jays were motivated to approach the more palatable food from the mirror panel more often compared to a less-palatable food, their motivation to engage and explore the mirror remained similar to what we observed in Task 1. The results of Task 3 also suggest that

birds might have perceived the mirror as a threatening stimulus to be avoided (whether a conspecific or not). Although there were no significant differences detected between conditions, only one jay passed the vertical mirror condition, whereas five passed each of the horizontal mirror and vertical wood conditions, and six passed the horizontal wood condition. The finding that fewer birds passed the vertical mirror condition compared with the horizontal mirror condition, even though they had never been presented with a horizontal mirror before but had had experience with the vertical mirror, suggests that birds might indeed interpret these two configurations differently. Grey parrots (*Pepperberg et al., 1995*), Goffin's cockatoos (*Van Buuren et al., 2018*), and jungle crows (*Kusayama, Bischof & Watanabe, 2000*), behaved more socially towards a vertical than a horizontal mirror. A vertical mirror seems more similar to a real-life bird, as birds are more often found upright, whereas they would only be standing on top of a bird, in any capacity, during agonistic or reproductive interactions. In our study, the horizontal mirror might have been perceived as less threatening than the vertical one. In the vertical configuration, the birds had to face the full view of their image in order to retrieve the food, in the horizontal configuration they could retrieve the food without stepping on the mirror and could see only a small portion of their face; this may also be more akin to a stimulus they would have observed naturally, such as when drinking from a large bowl of water (or in the wild, a lake or pond).

The behaviors displayed by the jays (absence of social behavior) during the mirror stimulation are quite different from corvid species tested so far using a similar set-up (*Prior, Schwarz & Güntürkün, 2008*; *Soler, Pérez-Contreras & Peralta-Sánchez, 2014*; *Van Buuren et al., 2018*; *Brecht, Müller & Nieder, 2020*; *Soler et al., 2020*; *Vanhooland, Bugnyar & Massen, 2020*; *Buniyaadi, Taufique & Kumar, 2020*). One possible explanation of what we observed, especially in Task 1, when the mirror and the wooden panel were first introduced, can be linked with a neophobic response. Although corvids are well known for their remarkable cognitive skills (*Taylor, 2014*; *Baciadonna et al., 2021b*), they are also highly neophobic (*Heinrich, 1995*; *O'Hara et al., 2017*). The combination of behavioral flexibility with high levels of neophobia appears paradoxical, because neophobia tends to inhibit innovation and is associated with narrow ecological niches (*Greggor, Thornton & Clayton, 2015*). In a recent paper (*Miller et al., 2022*) investigating the socio-ecological predictors of neophobia in ten different corvids species, Eurasian jays were considered the most neophobic towards novel food, and to some extent towards a novel object, compared with the rest of species included in the study. From the latency to approach the novel object reported by *Miller et al. (2022)*, it emerges that common ravens were more neophobic than azure-winged magpies, while azure-winged magpies, carrion crows, and Eurasian jays approached the novel objects with a similar, longer latency time than Clark's nutcrackers. This is quite interesting because these corvid species all failed to pass the classic mark-test and, more importantly, the behaviors often considered as indicators of the ability to pass the mark-test (exploratory behavior, contingent behavior and self-exploration) were limited both in the occurrences displayed, as well as in the number of individuals displaying them (*Brecht, Müller & Nieder, 2020*; *Vanhooland, Bugnyar & Massen, 2020*; *Wang et al., 2020*). Our results indicate that neophobia can posit a challenge for the jays, and most likely to

other corvids as well, to approach and explore the mirror, appreciate its reflective property, and use it to explore parts of their body otherwise not visible (*Vanhooland et al., 2023*).

However, the neophobic response cannot explain the observed behavior in Task 2 and 3 simply because the mirror element was not novel anymore. In addition, the behavior we observed, approaching the surface long enough to retrieve the reward and then flying away from it and to perches high in the compartment, suggest that the presence of palatable food was enough to motivate the birds to overcome their initial response towards the mirror but also strongly suggests that the motivation to explore the mirror played a marginal role (*Greenberg & Mettke-Hofmann, 2001*; *Miller et al., 2022*). A possible explanation for the observed lack of motivation to explore the mirror can be due to the fact that the tested birds needed even more exposure time, especially when assuming considerable high level of neophobia. For example, a recent study has shown a temporal dependence on the motivation to explore novel items (*O'Hara et al., 2017*). Species that were more neophobic (*e.g.*, slower to approach a novel item) compared to more neophilic species (*e.g.*, faster to approach a novel item) did not differ in terms of amount of exploration but differed in the onset of explorative behavior which occurred later in the neophobic animals. Although the total exposure time in our study is comparable to what has been previously found in corvids, the testing compartment where the jays were tested was larger than what has been used to test magpies, jackdaws and azure-winged magpies (*Prior, Schwarz & Güntürkün, 2008*; *Soler, Pérez-Contreras & Peralta-Sánchez, 2014*; *Soler et al., 2020*; *Wang et al., 2020*). Therefore, the time in direct view of the mirror was definitely less compared to when birds were tested in a more confined testing compartment and forced to face the mirror. The effective time in front of the mirror has also been proposed to explain the overall delayed occurrences of contingent behavior in ravens or a lack of interest towards the mirror in azure-winged magpies (*Vanhooland, Bugnyar & Massen, 2020*; *Vanhooland et al., 2023*). On the other hand, some suggestive evidence of MSR has been proposed to have been found in mirror-naïve Clark's nutcrackers presented with a social caching task in a small compartment (in which they were in view of a mirror most or all of the time; *Clary & Kelly, 2016*), as well as in another mirror-related cache-protection study involving California scrub-jays without previous mirror exposure (*Dally, Emery & Clayton, 2010*). However, we still cannot exclude that giving more opportunity to face the mirror might encourage jays to start to explore and interact with the mirror.

We also found that in Task 1 males took more food and looked at the surfaces more frequently compared to females. This result is quite interesting because, to our knowledge, sex differences in response to the mirror stimulation has never been examined before in corvids, often because the sex of the tested birds is unknown. Although further investigations are required to confirm our finding, the difference between males and females can be explained from an ecological perspective. Male jays are often dominant towards the female, they are involved in nest defense and during the breeding season males are motivated to feed their partner (*Goodwin, 1951*; *Goodwin, 1956*). These differences in behavioral strategy used by males and females may explain why males were more motivated to retrieve the food and glaze at the two panels placed in their compartment.

In our study, due to a general lack of motivation to explore the mirror, firm conclusions cannot be drawn, and further investigations are needed to assess the level of understanding of reflective surface by Eurasian jays and more importantly whether they saw their reflection as a conspecific or as themselves. In either case, it is appropriate to consider the natural and behavioral constraints of birds when attempting to administer mirror tasks to them, especially when tasks were originally conceived for apes. Limitations such as sensory preferences, physical constraints, neophobia, and more must be taken into account both when designing mirror tasks and when interpreting their results. Currently existing MSR tasks may pose particularly steep challenges for avian subjects. Eventually, a more conclusive understanding of the presence and extent of MSR in non-human animals, and especially in birds, is likely to only be attainable through continuous creativity and innovation in task design, rather than continuity of methodology.

## ACKNOWLEDGEMENTS

We are grateful to Maddalena Marengo for conducting a second video coding.

### Funding
This research was funded by the Templeton World Charity Foundation (TWCF0317), awarded to Nathan J. Emery and Nicola S. Clayton (funding Luigi Baciadonna, Nicola S. Clayton and Nathan J. Emery), as well as by a Herchel Smith Postgraduate Fellowship from Harvard University awarded to Francesca M. Cornero. The funders had no role in study design, data collection and analysis, decision to publish, or preparation of the manuscript.

### Grant Disclosures
The following grant information was disclosed by the authors:
Templeton World Charity Foundation: TWCF0317.
Herchel Smith Postgraduate Fellowship from Harvard University.

### Competing Interests
The authors declare there are no competing interests.

### Author Contributions
- Luigi Baciadonna conceived and designed the experiments, performed the experiments, analyzed the data, prepared figures and/or tables, authored or reviewed drafts of the article, and approved the final draft.
- Francesca M. Cornero performed the experiments, analyzed the data, authored or reviewed drafts of the article, and approved the final draft.
- Nicola S. Clayton conceived and designed the experiments, authored or reviewed drafts of the article, and approved the final draft.
- Nathan J. Emery conceived and designed the experiments, authored or reviewed drafts of the article, and approved the final draft.

## Animal Ethics

The following information was supplied relating to ethical approvals (*i.e.*, approving body and any reference numbers):

All experiments were approved by the University of Cambridge (ZOO63/19) and followed Home Office Regulations and the ASAB's Guidelines for the Treatment of Animals in Behavioural Research and Teaching.

## Data Availability

The raw data are available in the Supplementary File.

## Supplemental Information

Supplemental information for this article can be found online at http://dx.doi.org/10.7717/peerj.14729#supplemental-information.

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
