# Peer review of "Mirror stimulation in Eurasian jays (Garrulus glandarius)"

_PeerJ, doi:10.7717/peerj.14729_

## Round 0.1 · original submission · Major Revisions

Apologies for the slight delay in getting this decision to you. The reviews all came in while I was travelling and I needed to take the time to read the paper very carefully myself. I won’t reiterate the reviewers’ many helpful specific comments but please do be sure to address these thoroughly in the revision.

In summary, Reviewer 3 has some concerns about the novelty of the findings and the lack of clarity regarding the main aims of the study. All three reviewers are also concerned with the subjects’ prior experience with mirrors, something I am a bit more compelled to look past given that reflective surfaces are common in most animals’ environments. However, I do share reviewer 3’s concern with the tendency to accept a preferred interpretation of the data at the expense of other equally plausible interpretations. For example, this reviewer points to several places where motivation could explain the findings. I also agree with this reviewer’s questioning of whether the birds are really exhibiting fear in the absence of corroborating behavioral evidence. I also agree with this reviewer that the lengthy discussion of the MSR test detracts from what you actually studied here. I was surprised when, on line 121, you describe the tasks you presented and none seemed to explicitly involve self-recognition. It became more clear in the discussion what the aims might have been but these need to come first. Reviewer 1 notes that you have neglected some of the critical debates surrounding the MSR task; however, I think you could simply refer to some of these papers, state that MSR is controversial and then focus on other aspects of behavior impacted by mirrors. Although, in the discussion, there are many places where you should cite the work of Plotnik and others who have outlined necessary stages to precede formal mark tests. If this were my paper, I would introduce the tasks as novel ways to explore possible precursors to MSR or natural responses to mirrors without invoking MSR as a way to avoid imposing a known human ability on to other species for which it may not be appropriate. I do agree with Reviewer 3 that it is a bit unclear as to the true goal of the paper. Reviewer 2 noticed many errors or omissions that need to be corrected. We all found the paper to be well-written, but it needs to be tightened and focused on ideas relevant to what you tested.

I have a few comments of my own:

Lines 131-132, could a preference for the mirror reflect a social preference if the birds fail to understand that the reflection is of themselves?

Line 141, what is the rationale for investigating effects of orientation? This again is addressed finally in the discussion but not provided as a rationale at the outset. More careful control tests could have gotten at some of the issues you address in the discussion (e.g., viewing the face or not).

The way you describe the tasks from line 174 on makes it difficult to determine whether both surfaces were present and the bird was allowed to take a food from only one or whether only a single surface was presented on different trials. I gather the latter because of the wording around lines 205-6 but it needs to be less ambiguous. Why not counterbalance the order of all four conditions in Phase 3?

It would have been ideal to have a social condition where another bird was present behind a glass window so as to tease apart social from mirror effects. At the very least, it would have been helpful to have a condition that involved movement (but not a reflection). It is not surprising that birds look more at a stimulus that involves movement versus a static background.

Please report AIC values for your models.

You analyze sex but did not state hypotheses regarding sex differences. Please ensure all predictors and outcomes are justified by your background and rationale.

Line 388 replace “driver” with “predictor”

Reviewer 1 ·

Basic reporting

The manuscript is well written and quite easy to follow. I enjoy reading it and I think that it provides a step forward in understanding animals’ abilities to MSR and mirror-related behaviours.
However, I think there are some critical points.
In the introduction, authors gave only a little view on the literature available about the argument. For example, outside apes, they only mention MSR in elephants. There are many other species tested, such as dolphins (Reiss and Marino, 2001), horses (Baragli et al., 2021), giant pandas (Ma et al. 2015), and also fishes (Kohda et al., 2019).
In detail, I was astonished by reading autors’ claims at the end of the abstract and discussion:
- “Furthermore, we discuss how motivational factors and species differences must be taken into account when interpreting behavioural responses to the mirrors. In conclusion, after 50 years of intense MSR research, we believe it is necessary to develop new fruitful methods to test it, and change the way we approach MSR studies across species.”
- “Limitations such as sensory preferences, physical constraints, neophobia, and more must be taken into account both when designing mirror tasks and when interpreting their results, and currently existing MSR tasks may pose particularly steep challenges for avian subjects. Eventually, a more conclusive understanding of the presence and extent of MSR in non-human animals, and especially in birds, is likely to only be attainable through continuous creativity and innovation in task design, rather than continuity of methodology.”
What puzzled me is that the authors seem to have completely ignored the heated debate that has been going on about this topic recently. In 2021, Gallup and Anderson published a quite arrogant commentary about MSR in horses (Gallup GG, Anderson JR 2021 Putting the cart before the horse: claims for mirror self-recognition in horses are unfounded. Anim Cogn https:// doi. org/ 10. 1007/ s10071- 021- 01538-9) which was followed by a reply (Scopa et al., 2022) that states exactly the same things authors have stated in their abstract and discussion. I cannot explain why the authors have ignored such a vast and recent body of work on the subject that they themselves deal with.

Line 164-166 (Mat e Met): "They had also had exposure to mirrors during previous mirror-stimulation experiments: these included the placement of a mirror inside their aviary for two weeks and a mirror-guided string-pulling study (Baciadonna et al., 2021a)". This affirmation needs to be discussed. These bird were not naïve to mirror and authors totally neglect this fact in their discussion. I think this should be discussed. Being naïve to a mirror should be (I think) the fundamental step of every MSR test, however in some studies on MSR, animals were not completely naive to reflective surfaces, but they were tested anyway (chimpanzees, Gallup 1970; dolphins, Reiss and Marino 2001). Authors should discuss this fact and I think they also have to enlarge their introduction and discussion, taking into account the recent debate about it.
Moreover, in the discussion (399-401) authors said: "In our study, the jays tested were all adults (13-14 years old) that had not been exposed to mirrors at a young age, and we cannot rule out that a different behavioural pattern may have emerged if they had been confronted with the mirror during their younger, more neophilic stage." This sentence is ambigous and unclear. The tested subject had not been exposed to mirrors at young age, but they had still been exposed to mirror before this test. I think authors' affirmation sounds a little bit pander.

Experimental design

Line 150-152: I am not sure if including Lisbon in the task 3 is the right choice, given that this subject performed only one task out of three. Authors should explain better why they chose to added that subject.
Line 189-194: Why authors applied cling-film (that needed to be removed to retrieve the food) only in the task 2?

In line 242 authors stated that they built Generalised Linear Mixed Models using the lme4 package. Than, in line 246-251, they said they used the GLMM TMD package due to the non-gaussian distribution of the response variables. In line 252 they only mention “GLMM”. Which package?
Moreover, I think that they should use the same package for building GLMM. There are several packages that can build models with gaussian, binomial or poisson error distribution. This will make results more homogeneous and clear.

Validity of the findings

no comment

Additional comments

Line 278 - "The fixed Sex factor" should be "The fixed factor Sex"

Line 335-337: "The limited social behaviours displayed towards the reflected image throughout this study strongly suggest that the jays did not perceive the stimulus in the mirror as a conspecific worth interacting with or even confronting."
This sentence should be more parsimonius. Why the limited social behaviours displayed should strongly suggest that subject did not perceive the stimulus as a conspecific? Are there any other interpretations?

·

Basic reporting

Two minor comments on the labels of table 8 and figure 1:

1. In the legend of Table 8: “Table 8. Summary of the jays’ performance to approach and retrieve the food during Task 3. The green colour indicates the instances in which the tested subject approached and retrieved at least once the waxworm during the 15 s allowed in each session. The yellow colour instead indicates the instances in which the tested subject did not approach and retrieve at least once the waxworm during the 15 s allowed in each session.” You here mention 15-second sessions, this appears inconsistent with your methods section in which you mention 15-minute sessions (l.223-225).

2. In the legend of Figure 1 you mention panels E to G. These panels are missing from the figure and refer to a task 4 not presented in the manuscript.

Experimental design

3. L.145-147: Maybe you could elaborate on the hypothesis and or predictions for this task? What would be the reasoning or implications behind the birds having a preference for either condition?

Validity of the findings

no comment

Additional comments

This is a well-designed study and very well-written manuscript, I only have a few additional minor comments:
4. L.169-172: Given that these jays had pre-experience with mirrors and had participated in tasks with mirrors, I would assume that they had reached a certain level of habituation to the mirror prior to these tasks. How would you explain the (renewed?) neophobic reaction in the tasks of this study? Or did you notice a certain avoidance behaviour in the previous exposures as well?

5. l.423 “These further challenge (…)”: To what do you refer to with “These”?

6. Supplemental material: In the „README“-sheet of the Raw Data file: I suggest the correction to “Number of sessions performed” in cell B8

Reviewer 3 ·

Basic reporting

Overall, the manuscript is well written, and it covers few sensitive issues regarding cognitive skills in corvids.

Experimental design

- Main concern regards a relevant experimental bias: in line 162-166 it is reported that same subjects had previously participated in different experiments, also involving exposure to mirrors. In literature regarding MSR, regardless of the capacity to pass the mark test, experimental subjects must be naïve to mirrors. This is a conditio sine qua non to get reliable data on all those behaviors considered mandatory to be eligible for the mark test. This is also the reason why many studies about MSR have been criticized. This experimental bias inevitably leads to other concerns (see in Validity of findings).
- Line 135: I would suggest deepening this hypothesis according to which the presence of more-palatable food could increase exploratory behaviors towards a different thing which is not the food. Moreover, in line 189, authors also stated that this particularly palatable food was also covered with a film that needed to be removed by birds.
- For task 1 the jays had 16 fifteen-minute sessions; for task 2 jays had eight fifteen-minute sessions; for task 3 jays had first to successfully retrieved a worm from a tube 20 times, and then they can face the vertical/horizant mirros or wooden panel. However, no indications of how many sessions of this last task have been performed have reported. The criterion by which time duration of each session and number of sessions for each task have been defined, should be added.
- Regarding behaviors considered: exploration consists in food taken and also occurrences of looking. Since I supposed that looking displays are directed to the mirror or to the wooden panel and not the food, when it comes to consider exploration as a proxy behavior for passing the mark-test, exploration of food should not be taken into account. Also, it is really important to indicate which social, contingent and self-directed behaviors have been considered (in Table 1, Romero performed 17 contingent behaviors in front of the mirror in task 1, it would be interesting knowing what he did).

Validity of the findings

- The main issue is the fine line between the concept of perceiving a stimulus as threatening and not showing interest in the stimulus. Authors are likely to equalize these two possible explanations. Instead, especially because all the Discussion is focused on neophobic behavior, it is of outmost importance to define and distinguish fear from unconcern. Same problem when authors assumed that a preference for eating the food in front of the mirror may indicate an interest in the reflective surface. In fact, this behavior could also indicate lack of interest and total indifference towards the reflective surface. Since these birds are not naïve to mirrors, it is hard to completely rule out the possibility that they are just not engaged with the presence of a known apparatus and; whether this is true, there is no need to call upon neophobia.
- Line 363-365: I would remove or at least re-formulate this sentence which can create confusion instead of clarifying a not supported hypothesis. It should be pointed out that, assuming MSR, the perception of a conspecific must elicit the social response, which is one of the mandatory behaviors to be admitted to the mark test. Therefore, line 365 sounds improper. The capacity to “understand” that that conspecific is actually me, eventually appears after the social response. Also, this statement (linked to the previous one about dominance status) introduces another interesting issue, which is about the visual perception of the social status. Are there specific morphological features by which the dominance position of a jay can be visually perceived by another jay? Or it just depends on territoriality of the subjects? It could be worth deepening.
- Regarding the part about neophobia (till line 408): instead of explaining the lack of interest in the mirror with neophobia, I would investigate more the possibility of a lack of motivation, in terms of behavioral motive to approach the mirror. Especially when food is available wherever and when the mirror is no more a novelty.

Additional comments

- Abstract, line 32: MSR has been recently found also in horses (Baragli et al., 2021).
- Abstract, line 37-43: I would rephrase this sentence since it is based on MSR study. Nevertheless, this study is not about MSR. For this reason, I would address comments and discussion more on those behaviors which are indicative of the perception of the mirror surface, which when performed, are not necessarily related to the capacity of recognizing themselves.
- Keywords: I suggest removing “consciousness” and maybe add “neophobia”.
- Line 63: Remove “and some other non-human primates” since it is irrelevant here.
- Line 72: Baragli et al. 2021 broadly discussed this topic in their study as well. Also, around line 88, it is worth citing the study on horses which cannot use fingers to explore their body.
- Line 142: I suggest moving some information about different configurations between vertical and horizontal surfaces, from Discussion (line 356-358) to this paragraph, since it is not so easy to understand the choice of changing the shape of the panels.
- The main body of Introduction consists of questioning the validity of mark-test when applied on non-ape animals. Even though the issue is compelling and well-treated by authors, it is slightly off-topic since the study is not about MSR or mark-test. It would be more useful and interesting to review all the literature regarding the use of mirror, since many studies investigated this topic in mammals and birds. This approach would also be more coherent with the gradualist approach assumed, consisting of not being limited only to mark test when investigating cognitive skills in animals and their capacities to be aware of their own body or environment.

---

## Round 0.2 · Minor Revisions

Thank you for your careful revision and thoughtful responses to the comments from me and the reviewers. I am inviting another minor revision to address a few remaining issues identified by Reviewers 2 and 3. I have some very minor comments again myself:

All line numbers refer to the tracked change Word document.

You indicated that you did not have hypotheses regarding sex and aviary. This is fine but please still make it clear that they will be included in the model just to control for potential differences.

Please use American spelling (e.g., behavior) throughout.

There are two “which” on line 30.

Lines 33-34 move “only” to “in only a few species…” Check other misplacements of “only” (e.g., lines 88, 430)

I would delete “despite corvids being known..” on line 35. Excelling on one type of cognitive task does not imply the ability to excel in other domains. Similarly, I don’t find the description of results on lines 147-153 to present a compelling case for expecting jays to exhibit self-recognition. Unless you can make a compelling case for why these other tasks might make use of self-recognition or why self-recognition may share foundational abilities with these other capacities, I would be inclined to provide a stronger rationale for why jays might exhibit self-recognition.

Line 110, “insert “as” after “such.”

Lines 165-166 are a bit unclear, particularly the phrase, “with only one food type and thus of equal palatability.” Clarify if there are two equivalent pieces of food; one in front of each surface or whether there is one piece of food present elsewhere that can be eaten in front of either of these surfaces.

Line 409, ‘threating’ should be ‘threatening.’

The paragraph beginning with line 433 is too long. Start a new paragraph with “However” on line 458.

Line 489, change “has been never” to “has never been.”

Italicize all species names in the References.

Reviewer 1 ·

Basic reporting

Authors did a great and deep revision of their manuscript, addressing almost all the requests and softening their conclusions.

- Line 30, "which which" typo.

Experimental design

no comment

Validity of the findings

no comment

·

Basic reporting

No comment

Experimental design

No comment

Validity of the findings

No comment

Additional comments

The authors present here a carefully revised version of their manuscript “Mirror stimulation in Eurasian jays (Garrulus glandarius)” in which previous comments have been taken into account, resulting in an improved manuscript.
Please find below, my overall minor comments to this revised manuscript.

Abstract
Minor comment:
l.30 “which” is repeated twice in “Mirror exposure elicits a wide range of behavioural responses, some of which which have been considered as part“ I would suggest: Mirror exposure elicits a wide range of behavioural responses, some of which have been considered as part

l.33-34. In the sentence „Evidence of MSR has only been found categorically in few species, such as corvids, great apes, horses, and elephants“, you mix species and bigger taxonomic groups in your listing of species and the sentence leads a bit to the false assumption that MSR is quite commonly found in corvids, which is not the case. I would suggest to slightly rephrase the sentence.

Introduction:
Minor comment:
l.61. Would this apply to chimpanzees in particular or great apes in general?

l.62. Mentioning primates in general (and not for example great apes in particular here) might be a bit misleading, as beyond some claims in Rhesus Macaques, primate species outside of the great ape species do not seem to show “behavioural patterns in line with passing the test”.

l.108. There is one superfluous opening parenthesis in the reference

l.76-78. I would suggest a bit of caution pooling contingent and self-directed behaviours together here. If I recall correctly only self-directed behaviours were found to correlate with passing a MSR test in chimpanzees but not so contingent behaviours. Contingent behaviours do appear as a necessary step towards MSR but do not necessarily seem to indicated self-recognition.

Results:
General comment:
Could you please clarify whether the AIC value you report in your results section is the AIC of the full model or the delta AIC? From the way it is reported it reads like the AIC of the full model, yet from my understanding, AICs of single models provide little information if not considered in comparison with the AICs of other models (here the null model).

Discussion:
l.425.426: Birds do mount each other during copulation; however, the perspective of a male mounting the female would be a different from what they would see in a horizontal mirror as the male would see the females back and not her underside/belly.

Minor comment:
l.401-402: The phasing “mirror-guided self-exploration and self-directed behaviours” feels repetitive, I would suggest only keeping one of these terms

l.451-453: I would suggest rephrasing the sentence “the proxy behaviours often considered crucial for passing the mark-test (exploratory behaviour, contingent behaviour and self-exploration) were limited both in the occurrences displayed, as well as in the number of individuals displaying them”, as I would argue that exploratory behaviours and contingent behaviours are not per se essential for passing the mark test.

Reviewer 3 ·

Basic reporting

All suggestions have been taken into account by the authors. The manuscript addresses the issue of MSR in jays in a softer approach, by considering many other important aspects regarding the species-specific motivation, ecology, previous results.

Experimental design

- Linee 178: was the cling film covering the food present on both bowls (more-palatble food and normal daily food)? It can be worth adding this info here. Also, the reason why authors covered the food with the film is not explained.
- Line 185: Task 3 is described as food has been introduced in a position in which jays can land and collect it. Since the other two tasks were just described as food already present in the area, am I right supposing that the experimental paradigm was different in the third case? The reader may understand that in task 1 and 2 jays are presented with food in some sort of cage and in task 3 they have the opportunity to fly in front of two different configurations of a mirror (vertical or horizontal), land and collect the food and then fly again (which is in fact consistent with low interest in the reflective surfaces). If it is right, it is reasonable to think that jays do not have the occasion to appreciate the difference between vertical and horizontal configurations until not landing in front of them. So, how authors can be sure that birds actively chose a configuration instead of not landing by chance in front one of the two mirrors to collect the reward? The answer is in Material and Methods section, where readers can find out that in all the three tasks jays were inside an indoor testing compartment. Therefore, I suggest to remove the sentence about landing.

Validity of the findings

- Line 496: I agree with this interpretation. More interest of males towards the mirror can be just due to the fact that they are considered “bolder” than females, because of their social roles. Therefore, it is not interest related to the mirrors themselves since the same result can be potentially found related to any other new stimulus presented to male and female jays.

Additional comments

- Line 30: delate one “which”
- Introduction: all the suggestions have been considered. In the Introduction authors expose the main studies concerning use of mirrors in birds

---

## Round 0.3 · accepted · Accept

Thank you again for your attention to these final minor edits. I am pleased to accept your paper for publication in PeerJ.